# Corrosion Properties of Cr_27_Fe_24_Co_18_Ni_26_Nb_5_ Alloy in 1 N Sulfuric Acid and 1 N Hydrochloric Acid Solutions

**DOI:** 10.3390/ma14205924

**Published:** 2021-10-09

**Authors:** Chun-Huei Tsau, Po-Min Chen

**Affiliations:** Institute of Nanomaterials, Chinese Culture University, Taipei 111, Taiwan; stevenchen8900@gmail.com

**Keywords:** high-entropy alloy, Cr_27_Fe_24_Co_18_Ni_26_Nb_5_, microstructure, hardness, corrosion

## Abstract

The composition of the Cr_27_Fe_24_Co_18_Ni_26_Nb_5_ high-entropy alloy was selected from the FCC phase in a CrFeCoNiNb alloy. The alloy was melted in an argon atmosphere arc-furnace, followed by annealing in an air furnace. The dendrites of the alloy were in the FCC phase, and the eutectic interdendrites of the alloy comprised HCP and FCC phases. The microstructures and hardness of this alloy were examined; the results indicated that this alloy was very stable. This microstructure and hardness of the alloy almost remained the same after annealing at 1000 °C for 24 h. The polarization behaviors of Cr_27_Fe_24_Co_18_Ni_26_Nb_5_ alloy in 1 N sulfuric acid and 1 N hydrochloric acid solutions were measured. Both the corrosion potential and the corrosion current density of the Cr_27_Fe_24_Co_18_Ni_26_Nb_5_ alloy increased with increasing test temperatures. The activation energies of the Cr_27_Fe_24_Co_18_Ni_26_Nb_5_ alloy in these two solutions were also calculated.

## 1. Introduction

The high-entropy alloy concept [1,2] is now well known for designing new alloys. The four major characteristics of the high-entropy alloy concept are high entropy, sluggish diffusion, severe lattice distortion, and cocktail effects [3]. Materials researchers use this concept to smartly select appropriate elements to develop alloys for application fields [4]. This high-entropy alloy concept is also used to improve the corrosion resistance of metals, which is a very important issue. For example, the Cu_0.5_NiAlCoCrFeSi alloy had lower corrosion current densities than 304 stainless steel in H_2_SO_4_ and NaCl solutions [5], but the pitting potential of Cu_0.5_NiAlCoCrFeSi alloy was lower than that of 304 stainless steel in a NaCl solution. The CoCrFeNiTi alloy had a better pitting-corrosion potential than conventional corrosion-resistant alloys, such as Ni-based superalloys duplex stainless steels, in corrosion environments [6]. The FeCoNiNb and FeCoNiNb_0.5_Mo_0.5_ alloys had dual-phased dendritic microstructures, but they exhibited good corrosion resistance in comparison to 304 stainless steel in 1 M nitric acid and 1 M NaCl solutions [7]. The CrFeCoNiSn alloy also exhibited a good corrosion resistance with respect to 304 stainless steel in 0.6 M NaCl solution [8]. Moreover, the Al*_x_*CoCrFeNi (*x* = 0.15 and 0.4) high-entropy alloys exhibited better thermal stability and corrosion resistance compared to HR3C steel in a high-temperature and high-pressure environment [9]. The passivating elements, such as chromium, nickel, and molybdenum, are usually selected to be the elements of coating materials. Laser cladding, sputter deposition, and electro-spark deposition are frequently used processes. Al_0.5_CoCrCuFeNi alloy coating the surface of AZ91D by laser cladding successfully improved the corrosion resistance in 3.5 wt.% sodium chloride solution [10]. NbTiAlSiZrN_x_ alloy thin films sputtered on 304 stainless steel exhibited good corrosion resistance in 1 N H_2_SO_4_ solution [11]. Coatings of AlCr*_x_*NiCu_0.5_Mo (*x* = 0, 0.5, 1.0, 1.5, 2.0) alloys significantly improved the corrosion resistance of Q235 steel in 3.5% NaCl solution and a salt spray corrosion environment [12]. Cold working and annealing could also influence the electrochemical properties of 316 stainless steel, leading to an increase and a decrease in breakdown potential, respectively [13]. Shi et al. summarized the corrosion-resistant properties of high-entropy alloys in different solutions and corrosion environments and pointed out the methods for improving corrosion resistance [14].

In our previous study on CrFeCoNiNb alloys [15], the microstructures and corrosion behaviors of a CrFeCoNiNb alloy were investigated. The composition of Cr_27_Fe_24_Co_18_Ni_26_Nb_5_ alloy was selected from the FCC phase of the CrFeCoNiNb alloy. The microstructures, annealing effect, and polarization behaviors of the Cr_27_Fe_24_Co_18_Ni_26_Nb_5_ alloy were investigated in the present work.

## 2. Materials and Methods

The nominal compositions of Cr_27_Fe_24_Co_18_Ni_26_Nb_5_ alloy in weight percentage were Cr 24.22%, Fe 23.13%, Co 18.30%, Ni 26.34%, and Nb 8.01%. The alloy was melted in an arc-furnace in argon atmosphere. The total weight of the alloy was about 120 g. Part of the alloy was annealed at 1000 °C for different times. A scanning electron microscope was used to observe the microstructures of the alloy. An X-ray diffractometer was used to examine the crystal structures of the alloy; the scanning rate was 0.04°/s and the scanning range was 20–100°. A Vickers hardness tester was used to measure the hardness of the alloy; the loading force was 19.62 N (2 kg). The polarization behaviors of the alloy were measured using an electrochemical analyzer. A saturated silver chloride electrode (Ag/AgCl, SSE) was used as the reference electrode, and its potential was 0.197 V higher than that of the standard hydrogen electrode (SHE) at 25 °C [16]. A platinum wire was used as the counter electrode. The exposed area of the specimens was fixed at 0.1964 cm^2^ (diameter was 0.5 cm), and all of the specimens were wet-polished using 1200 grit SiC paper. The scanning rate of the polarization test was 0.001 V/s. Bubbling nitrogen gas was used to deaerate the oxygen in solutions during the polarization test. The polarization test was conducted for 900 s.

## 3. Results and Discussion

Figure 1 shows the micrographs of the Cr_27_Fe_24_Co_18_Ni_26_Nb_5_ alloy in as-cast and as-annealed states. The microstructures of the Cr_27_Fe_24_Co_18_Ni_26_Nb_5_ alloys were dendritic in both the as-cast and the as-annealed states. The dendrites of the as-cast Cr_27_Fe_24_Co_18_Ni_26_Nb_5_ alloy were in the FCC phase, whereas the interdendritic regions exhibited a dual-phased (FCC and HCP) eutectic structure, as shown in Figure 1a. The HCP phase was significantly spheroidized and coarsened after annealing, as shown in Figure 1b, resulting in a reduction in both the interphase area and the free energy of the alloy. Table 1 lists the chemical compositions of the overall, FCC, and HCP phases in the Cr_27_Fe_24_Co_18_Ni_26_Nb_5_ alloy in atomic percentage. The overall compositions of the Cr_27_Fe_24_Co_18_Ni_26_Nb_5_ alloy matched the theoretical values. The FCC phase in the Cr_27_Fe_24_Co_18_Ni_26_Nb_5_ alloy had less Nb and Co, but more Cr, Fe, and Ni. A possible reason was that the elements of niobium and cobalt potentially formed a melt with a low melting point because of a eutectic reaction, thus forming the HCP phase in the interdendrites of the alloy during casting, featuring more Nb and Co. The Co–Ni binary phase diagram [17] shows that the melting point of a Co–20.3Nb alloy is only 1237 °C.

The Cr_27_Fe_24_Co_18_Ni_26_Nb_5_ alloy was based on the compositions in the FCC phase of CrFeCoNiNb alloy [15]. However, the Cr_27_Fe_24_Co_18_Ni_26_Nb_5_ alloy was a dual-phased alloy, because the solid solubility of Nb in the alloy could change its composition. According to our previous studies on CrFeCoNiNb_x_ alloys [15,18], the Nb content in the FCC phase of CrFeCoNiNb*_x_* alloys changes with the Nb content(*x*), as shown in Figure 2. The Nb content in the FCC phase of the alloys increases with the Nb content, becoming almost saturated at *x* = 0.6. In the present study, the Nb content of the alloy was only 5 at.%; thus, its solid solubility in the FCC phase was reduced, again forming the HCP phase in the Cr_27_Fe_24_Co_18_Ni_26_Nb_5_ alloy. However, the Cr_27_Fe_24_Co_18_Ni_26_Nb_5_ alloy is a five-element alloy, not a binary alloy.

Figure 3 displays the XRD patterns of the Cr_27_Fe_24_Co_18_Ni_26_Nb_5_ alloy in as-cast and as-annealed states. Only two phases were detected: one was an FCC phase with a lattice constant of 3.58 Å, and the other was an HCP phase with lattice constants of 4.80 Å (*a*-axis) and 7.83 Å (*c*-axis). Heat treatment did not significantly influence the lattice constants and the relative intensities of these two phases. For example, the Cr_27_Fe_24_Co_18_Ni_26_Nb_5_ alloy was very stable, even though it was annealed at 1000 °C for 24 h. Figure 4 plots the relationship between the hardness of the Cr_27_Fe_24_Co_18_Ni_26_Nb_5_ alloy and the annealing time; the annealing temperature was 1000 °C. The hardness of the Cr_27_Fe_24_Co_18_Ni_26_Nb_5_ alloy remained at approximately 250 HV. This also proves that annealing at 1000 °C had almost no influence on this alloy. The volume fraction of the HCP phase was less than that of the FCC phase; therefore, the coarsening and spheroidizing of the HCP phase had no apparent effect on the hardness of the alloy. The HCP and FCC phases had different compositions and structures; however, the ratios of these two phases did not change significantly. Therefore, the as-cast Cr_27_Fe_24_Co_18_Ni_26_Nb_5_ alloy was selected to test its polarization behavior in 1 N H_2_SO_4_ and 1 N HCl solutions.

The polarization curves of the Cr_27_Fe_24_Co_18_Ni_26_Nb_5_ alloy tested in deaerated 1 N H_2_SO_4_ solution and 1 N HCl solution under different temperatures are shown in Figure 5a,b, respectively. The curve with a potential lower than the corrosion potential (*E*_corr_) represents the cathodic curve, whereby the alloy under this state would be protected; the curve with a potential higher than the corrosion potential represents anodic curve, whereby the alloy under this state would be corroded. The cathodic line of the Cr_27_Fe_24_Co_18_Ni_26_Nb_5_ alloy exhibited a Tafel slope (β_c_); β_c_ = Δ*E*/Δlog*i*, where *E* is the potential, and *i* is the current density. The current density corresponding to *E*_corr_ is the corrosion current density (*i*_corr_). The current density of the alloy increases with the applied potential (overvoltage) before decreasing upon passing the anodic peak and entering the passivation region. Figure 5a displays the polarization curves of the Cr_27_Fe_24_Co_18_Ni_26_Nb_5_ alloy tested in 1 N H_2_SO_4_ solution. The corrosion potentials and corrosion current densities increased with test temperature. Furthermore, the current densities of the anodic peaks (*i*_pp_) and passivation regions (*i*_pass_) increased with test temperature. However, all of the passivation regions of the Cr_27_Fe_24_Co_18_Ni_26_Nb_5_ alloy tested in deaerated 1 N H_2_SO_4_ solution retained complete shapes in the temperature range of 30–60 °C. The polarization data, namely, the Tafel slope (β_c_), corrosion potentials (*E*_corr_), corrosion current densities (*i*_corr_), passivation potential (*E*_pp_, potential of the anodic peak), anodic critical current density of the anodic peak (*i*_pp_), passive current density (*i*_pass_), and breakdown potential (*E*_b_), of the Cr_27_Fe_24_Co_18_Ni_26_Nb_5_ alloy tested in deaerated 1 N H_2_SO_4_ solution under different temperatures are listed in Table 2. The polarization curves of the Cr_27_Fe_24_Co_18_Ni_26_Nb_5_ alloy tested in deaerated 1 N HCl solution under different temperatures are shown in Figure 5b. The corrosion potentials and corrosion current densities of the Cr_27_Fe_24_Co_18_Ni_26_Nb_5_ alloy tested in deaerated 1 N HCl solution increased with test temperature, similar to the results of the alloy tested in deaerated 1 N H_2_SO_4_ solution. However, the anodic peaks of the Cr_27_Fe_24_Co_18_Ni_26_Nb_5_ alloy tested in deaerated 1 N HCl solution were larger than those of the alloy tested in 1 N H_2_SO_4_ solution. A large anodic peak indicates that the alloy was harder upon entering the passivation region in 1 N HCl solution. Moreover, breakdown of the passivation region of the Cr_27_Fe_24_Co_18_Ni_26_Nb_5_ alloy started at a testing temperature of 50 °C, becoming very clear at a testing temperature of 60 °C. This suggests that the Cr_27_Fe_24_Co_18_Ni_26_Nb_5_ alloy did not resist the attack from chloride ions at higher temperatures. The corrosion potentials (*E*_corr_) and corrosion current densities (*i*_corr_) of Cr_27_Fe_24_Co_18_Ni_26_Nb_5_ alloy tested in deaerated 1 N HCl solution under different temperatures are also listed in Table 2.

Figure 6 shows the Arrhenius plot of relationships between corrosion current density of the Cr_27_Fe_24_Co_18_Ni_26_Nb_5_ alloy and the test temperature in the two solutions. The relationship between corrosion current density and testing temperature satisfied the relationship of *i*_corr_ = Aexp(−Q/RT), where *i*_corr_ is the corrosion current density, A is a constant, Q is the activation energy, R is the gas constant, and T is the temperature. Therefore, the activation energy Q could be calculated by plotting ln*i*_corr_ vs. 1/T, as shown in Figure 6. The activation energies of corrosion of the Cr_27_Fe_24_Co_18_Ni_26_Nb_5_ alloy in 1 N H_2_SO_4_ solution and 1 N HCl solution were 27.7 and 52.9 kJ/mol, respectively. Thus, the Cr_27_Fe_24_Co_18_Ni_26_Nb_5_ alloy tested in 1 N HCl solution had a larger activation energy.

The morphologies of the Cr_27_Fe_24_Co_18_Ni_26_Nb_5_ alloy after polarization in these two solutions are shown in Figure 7; both cases showed a uniform corrosion morphology. Both the FCC and the HCP phases in the Cr_27_Fe_24_Co_18_Ni_26_Nb_5_ alloy were corroded after polarization in 1 N H_2_SO_4_ solution at 30 and 50 °C, as shown in Figure 7a,b, respectively. However, the FCC phase was severely more corroded than the HCP phase. Figure 7c,d show the morphologies of the Cr_27_Fe_24_Co_18_Ni_26_Nb_5_ alloy after polarization in 1 N HCl solution at 30 and 50 °C, respectively. The FCC phase was also severely corroded, but the HCP phase did not display any corrosion and kept its original shape. Therefore, the FCC phase was the major corroded phase of this alloy in both solutions.

## 4. Conclusions

The microstructure and corrosion behavior of the Cr_27_Fe_24_Co_18_Ni_26_Nb_5_ alloy were studied. This alloy had a dendritic structure with two phases, FCC and HCP. The dendrites were in the FCC phase, whereas the interdendrites exhibited a dual-phased eutectic structure. The Cr_27_Fe_24_Co_18_Ni_26_Nb_5_ alloy maintained its structure and hardness after annealing at 1000 °C for 24 h. All polarization curves of the Cr_27_Fe_24_Co_18_Ni_26_Nb_5_ alloy displayed complete shapes in the temperature range of 30–60 °C in 1 N H_2_SO_4_ solution, but a breakdown of the passivation region of the Cr_27_Fe_24_Co_18_Ni_26_Nb_5_ alloy in HCl solution was observed at 50 °C. The Cr_27_Fe_24_Co_18_Ni_26_Nb_5_ alloy showed uniform corrosion morphologies in both 1 N H_2_SO_4_ and 1 N HCl solutions. The corrosion activation energies of the Cr_27_Fe_24_Co_18_Ni_26_Nb_5_ alloy in 1 N H_2_SO_4_ and 1 N HCl solutions were 27.7 and 52.9 kJ/mol, respectively. However, the FCC phase in the Cr_27_Fe_24_Co_18_Ni_26_Nb_5_ alloy was severely more corroded than the HCP phase.

## Figures and Tables

**Figure 1 materials-14-05924-f001:**
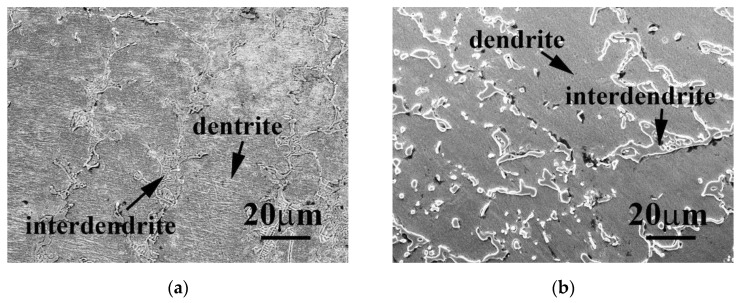
SEM micrographs of Cr_27_Fe_24_Co_18_Ni_26_Nb_5_ alloy in the (**a**) as-cast state, and (**b**) after annealing at 1000 °C for 24 h.

**Figure 2 materials-14-05924-f002:**
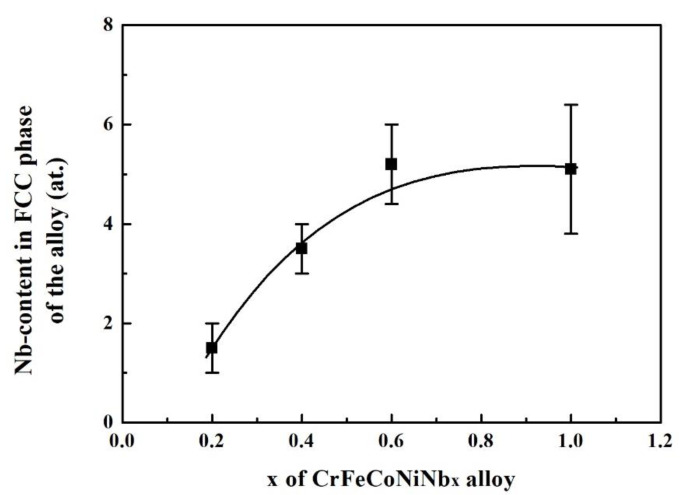
Plot of Nb content in FCC phase of a CrFeCoNiNb_x_ alloy as a function of Nb content [15,18].

**Figure 3 materials-14-05924-f003:**
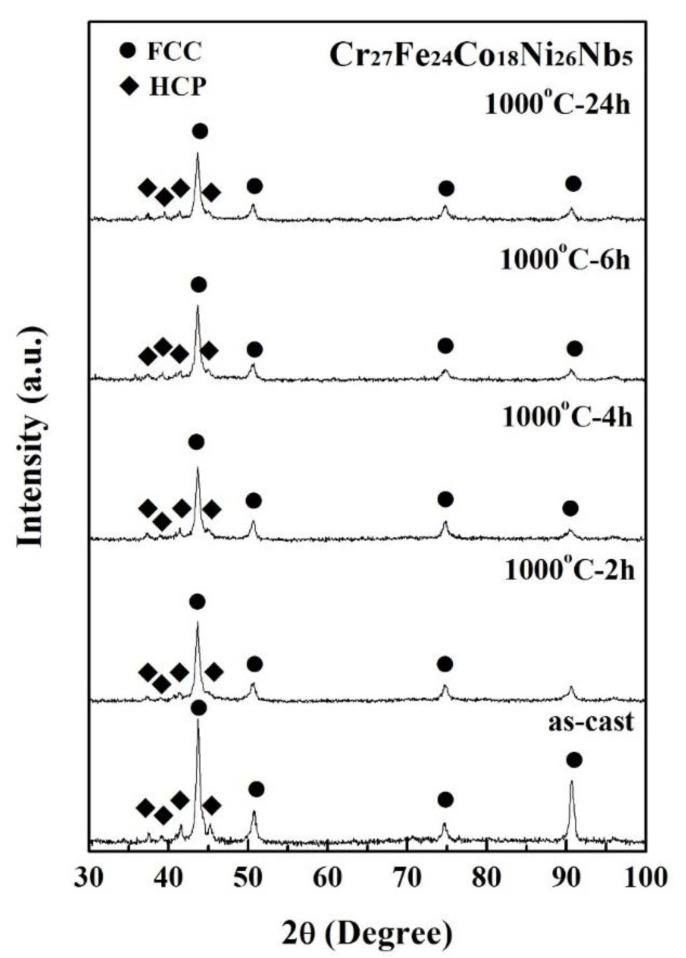
XRD patterns of Cr_27_Fe_24_Co_18_Ni_26_Nb_5_ alloy in as-cast and as-annealed conditions.

**Figure 4 materials-14-05924-f004:**
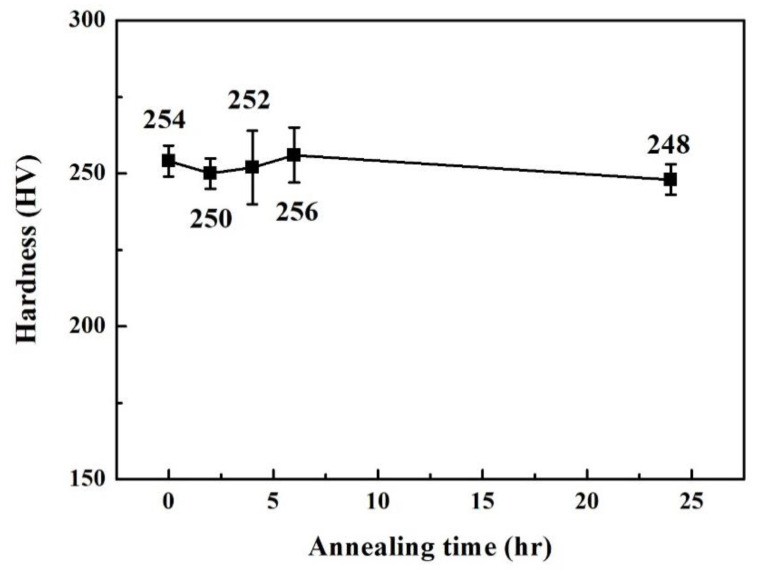
Plot of hardness of Cr_27_Fe_24_Co_18_Ni_26_Nb_5_ alloy as a function of annealing time; the annealing temperature was 1000 °C.

**Figure 5 materials-14-05924-f005:**
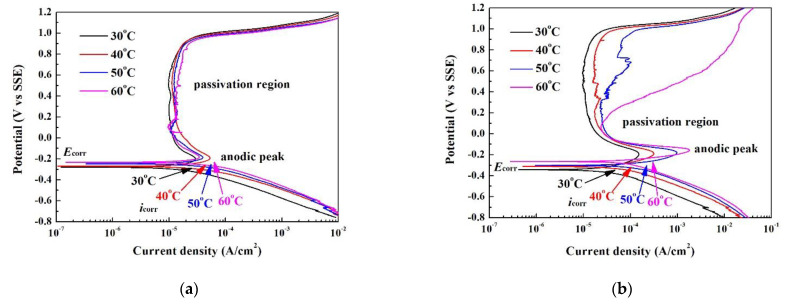
Polarization curves of the Cr_27_Fe_24_Co_18_Ni_26_Nb_5_ alloy tested in (**a**) deaerated 1 N H_2_SO_4_ solution; and (**b**) deaerated 1 N HCl solution under different temperatures.

**Figure 6 materials-14-05924-f006:**
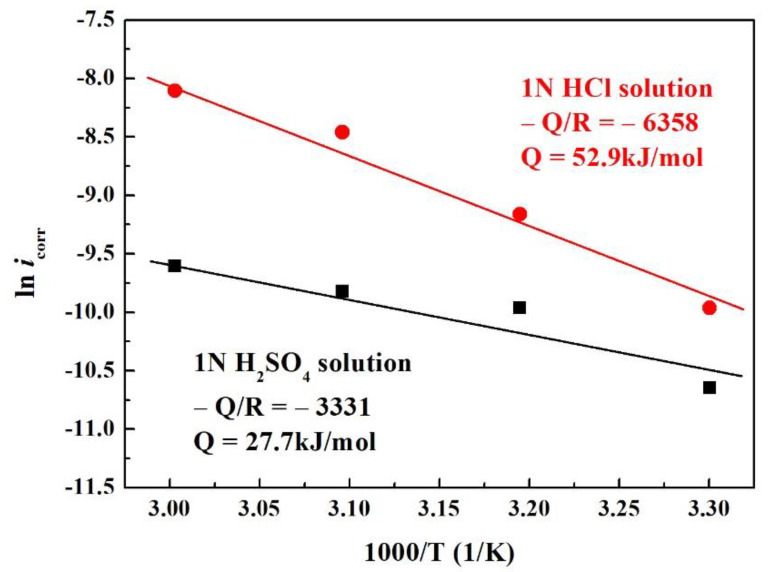
Arrhenius plot displaying the relationship between corrosion current density and testing temperature of the Cr_27_Fe_24_Co_18_Ni_26_Nb_5_ alloy tested in deaerated 1 N H_2_SO_4_ and deaerated 1 N HCl solutions.

**Figure 7 materials-14-05924-f007:**
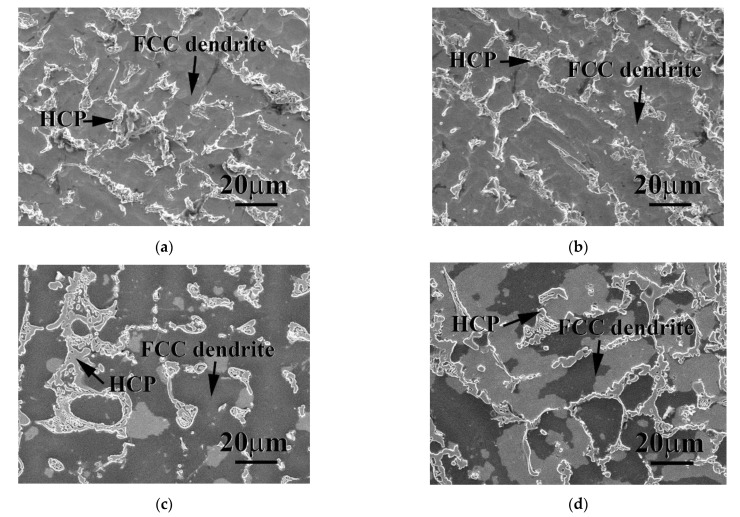
Morphologies of the Cr_27_Fe_24_Co_18_Ni_26_Nb_5_ alloy after polarization in (**a**) deaerated 1 N H_2_SO_4_ solution at 30 °C, (**b**) deaerated 1 N H_2_SO_4_ solution at 50 °C, (**c**) deaerated 1 N HCl solution at 30 °C, and (**d**) deaerated 1 N HCl solution at 50 °C.

**Table 1 materials-14-05924-t001:** Chemical compositions of the phases in the Cr_27_Fe_24_Co_18_Ni_26_Nb_5_ alloy analyzed by SEM/EDS in atomic percentage.

Phase	Cr	Fe	Co	Ni	Nb
Overall	26.0 ± 0.8	24.1 ± 0.9	18.1 ± 0.8	25.8 ± 0.7	6.0 ± 0.5
FCC	27.5 ± 0.5	25.7 ± 0.6	17.7 ± 1.0	26.5 ± 0.8	2.6 ± 0.5
HCP	18.1 ± 1.6	15.4 ± 2.1	26.4 ± 2.1	19.7 ± 2.8	30.4 ± 4.3

**Table 2 materials-14-05924-t002:** Polarization data of the Cr_27_Fe_24_Co_18_Ni_26_Nb_5_ alloy tested in deaerated 1 N H_2_SO_4_ and deaerated 1 N HCl solutions.

Solution	Items	30 °C	40 °C	50 °C	60 °C
1 N	**β_c_ (V·cm^2^/A)**	0.178	0.177	0.187	0.200
H_2_SO_4_	***E*_corr_ (V vs. SSE)**	−0.280	−0.272	−0.246	−0.233
solution	***i*_corr_ (μA/cm^2^)**	23.7	47.0	54.2	67.3
	***E*_pp_ (V vs. SSE)**	−0.200	−0.192	−0.186	−0.181
	***i*_pp_ (μA/cm^2^)**	29.3	52.9	39.4	34.3
	***i*_pass_ (μA/cm^2^)**	10.1	12.1	11.6	10.0
	***E*_b_ (V vs. SSE)**	0.981	0.972	0.960	0.955
1 N	**β_c_ (V·cm^2^/A)**	1.17	1.34	1.31	1.37
HCl	***E*_corr_ (V vs. SSE)**	−0.343	−0.312	−0.302	−0.265
solution	***i*_corr_ (μA/cm^2^)**	47.1	105	212	302
	***E*_pp_ (V vs. SSE)**	−0.196	−0.188	−0.180	−0.156
	***i*_pp_ (mA/cm^2^)**	0.152	0.315	0.971	1.82
	***i*_pass_ (μA/cm^2^)**	9.78	10.8	23.5	23.7
	***E*_b_ (V vs. SSE)**	1.009	1.001	0.994	0.233

## Data Availability

Not applicable.

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
