# Peer review of "Corrosion Properties of Cr27Fe24Co18Ni26Nb5 Alloy in 1 N Sulfuric Acid and 1 N Hydrochloric Acid Solutions"

_materials, 2021, doi:10.3390/ma14205924_

Round 1

Reviewer 1 Report

Dear authors. I have gone through your paper with great interest. Unfortunately, I find it in the current form unsuitable for publication. Please see my comments below:

  • The work lacks fluidity and more importantly, a deep discussion allowing the understanding of the results, which have been essentially described.
  • The authors should consider justifying the alloys selection concerning the corrosion resistance and the role of each constituent and fraction to enables the desirable microstructure. No justification of the selected electrolytes.
  • The main remark remains the lack of discussion; basically the work consist of the description of events, which do not characterize a scientific work.

Author Response

The work lacks fluidity and more importantly, a deep discussion allowing the understanding of the results, which have been essentially described.

Reply: We have modified our manuscript in accordance with all reviewers’ commands.

The authors should consider justifying the alloys selection concerning the corrosion resistance and the role of each constituent and fraction to enables the desirable microstructure. No justification of the selected electrolytes.

Reply: Both of “one alloy under different conditions” and “different alloys under one condition” are research methods. Three published paper which studied the corrosion properties of one alloy are listed:

Corrosion Behavior of Alloy C-276 in Supercritical Water, Advances in Materials Science and Engineering, 2018, https://doi.org/10.1155/2018/1027640

Corrosion Behavior of Al0.1CoCrFeNi High Entropy Alloy in Various Chloride-Containing Solutions, Frontiers in Materials, 2021, https://doi.org/10.3389/fmats.2020.533843

Study on Corrosion Resistance of N36 Zirconium Alloy in LiOH Aqueous Solution, World Journal of Nuclear Science and Technology, 2018, Vol.8 No.2

The main remark remains the lack of discussion; basically the work consist of the description of events, which do not characterize a scientific work.

Reply: We have enhanced our manuscript.

Reviewer 2 Report

This study investigated the corrosion properties of Cr27Fe24Co18Ni26Nb5 alloy using sulfuric acid and hydrochloric acid solutions.  Some revisions are suggested before publication.

1. Why is the hardness value maintained despite heat treatment at 1000 °C? It is expected that the grain size will be coarsened by the heat treatment. Finally, the ratio of FCC phase and HCP phase seems to have influenced the hardness value. There is no mention related to this content.

2. It is better to change the classification of chemical composition to dendrite/interdendrite in table 1. (FCC/HCP -> dendrite/interdendrite) 

3. What causes the corrosion potential and corrosion current density to tend to increase with increasing test temperature? Probably because the fraction of HCP phase increased due to the increase in test temperature. The relationship between microstructure fractions and polarization test results should be explained in detail.

4. Is the cause of the results shown in Figure 6 due to the corroded phase types? (H2SO4 solution - HCP, HCl solution - FCC) 

5. Why did you perform the heat treatment at 1000 °C? Is it because of the precipitation of the HCP phase? 

Author Response

  1. Why is the hardness value maintained despite heat treatment at 1000 °C? It is expected that the grain size will be coarsened by the heat treatment. Finally, the ratio of FCC phase and HCP phase seems to have influenced the hardness value. There is no mention related to this content.

Reply: We chose high temperature of 1000 °C to test the stability of this alloy. The volume fraction of HCP phase was less than that of FCC phase; therefore, the coarsening and spheroidizing of HCP phase had no apparent effect on the hardness of the alloy. The HCP and FCC phases had different compositions and structures; so that, the ratios of these two phase did not change significantly. These descriptions has been added into this manuscript.

  1. It is better to change the classification of chemical composition to dendrite/interdendrite in table 1. (FCC/HCP -> dendrite/interdendrite) 

Reply: The interdendritic regions was a eutectic structure, and it had two phases, HCP and FCC. Therefore, we measured the compositions of HCP and FCC phases by SEM/EDS.

  1. What causes the corrosion potential and corrosion current density to tend to increase with increasing test temperature? Probably because the fraction of HCP phase increased due to the increase in test temperature. The relationship between microstructure fractions and polarization test results should be explained in detail.

Reply: As same as chemical reaction, the reaction rate of electrochemical reaction increases with increasing test temperature. That is, the corrosion rate increases with increasing temperature, and corrosion current density increases because more atoms become ions and release electrons. To determine the variation of corrosion potential must measure the concentration of the ions of each elements, and then calculates the variation of potential by Nernst equation. This part is not included in our manuscript.

  1. Is the cause of the results shown in Figure 6 due to the corroded phase types? (H2SO4 solution - HCP, HCl solution - FCC) 

Reply: Figure 6 is the plot to calculate the activation energies of this alloy in these two solutions. The corrosion current densities were from the Figure 5 and Table 2. Arrhenius equation (or rate equation), rate=A exp(-Q/RT), dose not relate to the corrosion type.

  1. Why did you perform the heat treatment at 1000 °C? Is it because of the precipitation of the HCP phase? 

Reply: We want to test the stability of this alloy. The highest temperature of air furnace we used was 1100 °C. So we chose 1000 °C. The HCP phase was not precipitates, HCP phase always existed after casting.

Reviewer 3 Report

The novelty statement at the end of the manuscript is not sufficient and should be explained more.
Provide further evidence to detect the chemical composition of the different phases.
Give a complete numerical data table including the Tafel slope, breakdown potential, passivation currents (ipass), and anodic peaks currents (ipp) extracted from the diagrams in Figure 5.
An impedance test is necessary to determine the corrosion stages.
More microscopic images of the structure before and after corrosion at different temperatures should be provided.
The literature review is not sufficient and authors must review and cite more papers in the field and especially newly published ones. Doing this, reviewing the following refs could be helpful:
[a] Journal of Materials Engineering and Performance, 27, 2018, 271-281
[b] Journal of Alloys and Compounds 860, 2021, 158412
[c] Metals 7(2), 2017, 43

Author Response

The novelty statement at the end of the manuscript is not sufficient and should be explained more.

Reply: We have modified our manuscript from all reviewers’ commands.

Provide further evidence to detect the chemical composition of the different phases.

Reply: Some SEM/EDS figures are shown in attachment. We do not show these figures in this manuscript, because they need too many space of printing page.

Give a complete numerical data table including the Tafel slope, breakdown potential, passivation currents (ipass), and anodic peaks currents (ipp) extracted from the diagrams in Figure 5.

Reply: Those data were added into Table 2.

An impedance test is necessary to determine the corrosion stages.

Reply: The impedance test is not included in this manuscript. We will consider to do it in the future. Thank you.

More microscopic images of the structure before and after corrosion at different temperatures should be provided.

Reply: We have added the figures.

The literature review is not sufficient and authors must review and cite more papers in the field and especially newly published ones. Doing this, reviewing the following refs could be helpful:

[a] Journal of Materials Engineering and Performance, 27, 2018, 271-281
[b] Journal of Alloys and Compounds 860, 2021, 158412
[c] Metals 7(2), 2017, 43

Reply: These references are added into our manuscript.

Round 2

Reviewer 1 Report

The manuscript can be accepted in the present form.

Reviewer 2 Report

The author answered most of my comments. Therefore, I can accept this manuscript  now. Thanks.

Reviewer 3 Report

The revised manuscript could be considered for publication.